# Heart murmur detection from phonocardiogram recordings: The George B. Moody PhysioNet Challenge 2022

**Matthew A. Reyna**[1]*, **Yashar Kiarashi**[1], **Andoni Elola**[2], **Jorge Oliveira**[3], **Francesco Renna**[4], **Annie Gu**[1], **Erick A. Perez Alday**[1], **Nadi Sadr**[1,5], **Ashish Sharma**[1†], **Jacques Kpodonu**[6], **Sandra Mattos**[7], **Miguel T. Coimbra**[4], **Reza Sameni**[1], **Ali Bahrami Rad**[1], **Gari D. Clifford**[1,8]

**1** Department of Biomedical Informatics, Emory University, Atlanta, Georgia, United States of America, **2** Department of Electronic Technology, University of the Basque Country UPV/EHU, Eibar, Spain, **3** REMIT, Universidade Portucalense, Porto, Portugal, **4** INESC TEC, Faculdade de Ciências, Universidade do Porto, Porto, Portugal, **5** ResMed, Sydney, Australia, **6** Division of Cardiac Surgery, Beth Israel Deaconess Medical Center, Harvard Medical School, Boston, Massachusetts, United States of America, **7** Unidade de Cardiologia e Medicina Fetal, Real Hospital Português, Recife, Brazil, **8** Department of Biomedical Engineering, Emory University and the Georgia Institute of Technology, Atlanta, Georgia, United States of America

† Deceased.
¤ Current address: Department of Biomedical Informatics, Emory University, Atlanta, Georgia, United States
* matthew.a.reyna@emory.edu

**Data Availability Statement:** J. H. Oliveira, F. Renna, P. Costa, D. Nogueira, C. Oliveira, C. Ferreira, A. Jorge, S. Mattos, T. Hatem, T. Tavares, A. Elola, A. Rad, R. Sameni, G. D. Clifford, & M. T. Coimbra (2021). The CirCor DigiScope Dataset: From Murmur Detection to Murmur Classification. IEEE Journal of Biomedical and Health Informatics,

## Abstract

Cardiac auscultation is an accessible diagnostic screening tool that can help to identify patients with heart murmurs, who may need follow-up diagnostic screening and treatment for abnormal cardiac function. However, experts are needed to interpret the heart sounds, limiting the accessibility of cardiac auscultation in resource-constrained environments. Therefore, the George B. Moody PhysioNet Challenge 2022 invited teams to develop algorithmic approaches for detecting heart murmurs and abnormal cardiac function from phonocardiogram (PCG) recordings of heart sounds. For the Challenge, we sourced 5272 PCG recordings from 1452 primarily pediatric patients in rural Brazil, and we invited teams to implement diagnostic screening algorithms for detecting heart murmurs and abnormal cardiac function from the recordings. We required the participants to submit the complete training and inference code for their algorithms, improving the transparency, reproducibility, and utility of their work. We also devised an evaluation metric that considered the costs of screening, diagnosis, misdiagnosis, and treatment, allowing us to investigate the benefits of algorithmic diagnostic screening and facilitate the development of more clinically relevant algorithms. We received 779 algorithms from 87 teams during the Challenge, resulting in 53 working codebases for detecting heart murmurs and abnormal cardiac function from PCG recordings. These algorithms represent a diversity of approaches from both academia and industry, including methods that use more traditional machine learning techniques with engineered clinical and statistical features as well as methods that rely primarily on deep learning models to discover informative features. The use of heart sound recordings for identifying heart murmurs and abnormal cardiac function allowed us to explore the potential

https://doi.org/10.1109/JBHI.2021.3137048. The Challenge data can be downloaded from Oliveira, J., Renna, F., Costa, P., Nogueira, M., Oliveira, A. C., Elola, A., Ferreira, C., Jorge, A., Bahrami Rad, A., Reyna, M., Sameni, R., Clifford, G., & Coimbra, M. (2022). The CirCor DigiScope Phonocardiogram Dataset (version 1.0.3). PhysioNet. https://doi.org/10.13026/tshs-mw03.

**Funding:** This research is supported by the National Institute of General Medical Sciences (NIGMS) and the National Institute of Biomedical Imaging and Bioengineering (NIBIB) under NIH grant numbers 2R01GM104987-09 and R01EB030362 respectively, the National Center for Advancing Translational Sciences of the National Institutes of Health under Award Number UL1TR002378, as well as the Gordon and Betty Moore Foundation and MathWorks under unrestricted gifts. AE receives financial support through grant PID2021-122727OB-I00 funded by MCIN/AEI/10.13039/501100011033 and "ERDF A way of making Europe" and by the Basque Government under Grant IT1717-22. FR and MC receive financial support by National Funds through the Portuguese funding agency, FCT - Fundação para a Ciência e a Tecnologia, within project UIDB/50014/2020. None of the aforementioned entities influenced the design of the Challenge or provided data for the Challenge. The funders had no role in study design, data collection and analysis, decision to publish, or preparation of the manuscript.

**Competing interests:** I have read the journal's policy and the authors of this manuscript have the following competing interests: AE receives financial support through grant PID2021-122727OB-I00 funded by MCIN/AEI/10.13039/501100011033 and "ERDF A way of making Europe" and by the Basque Government under Grant IT1717-22. FR and MC receive financial support by National Funds through the Portuguese funding agency, FCT - Fundação para a Ciência e a Tecnologia, within project UIDB/50014/2020. GC has financial interests in AliveCor, LifeBell AI and Mindchild Medical. GC also holds a board position in LifeBell AI and Mindchild Medical.

of algorithmic approaches for providing more accessible diagnostic screening in resource-constrained environments. The submission of working, open-source algorithms and the use of novel evaluation metrics supported the reproducibility, generalizability, and clinical relevance of the research from the Challenge.

## Author summary

Cardiac auscultation is an accessible diagnostic screening tool for identifying heart murmurs. However, experts are needed to interpret heart sounds, limiting the accessibility of auscultation. Therefore, the George B. Moody PhysioNet Challenge 2022 invited teams to develop algorithms for detecting heart murmurs and abnormal cardiac function from phonocardiogram (PCG) recordings of heart sounds. For the Challenge, we sourced 5272 PCG recordings from 1452 primarily pediatric patients in rural Brazil. We required the participants to submit the complete code for their algorithms, improving the transparency, reproducibility, and utility of their work. We also devised an evaluation metric that considered the costs of screening, diagnosis, misdiagnosis, and treatment, allowing us to investigate the benefits of algorithmic diagnostic screening and facilitate the development of more clinically relevant algorithms. We received 779 algorithms from 87 teams during the Challenge, resulting in 53 working codebases and publications that represent a diversity of algorithmic approaches to detecting heart murmurs and identifying clinical outcomes from heart sound recordings.

## Introduction

Cardiac auscultation via stethoscopes remains the most common and the most cost-effective tool for cardiac pre-screening. However, despite its popularity, cardiac auscultation has limited diagnostic sensitivity and accuracy because the interpretation of heart sounds from stethoscopes requires many years of experience and skill, leading to significant disagreement between medical personnel, [2, 3, 4]. Digital phonocardiography has emerged as a more objective alternative to traditional cardiac auscultation. The phonocardiogram (PCG) can be acquired by a combination of high-fidelity stethoscope front-ends and high-resolution digital sampling circuitry, which enable the registration of heart sounds as a discrete-time signal and enable the use of algorithmic methods for heart sound analysis and diagnosis [5].

As acoustic signals, heart sounds are mainly generated by the vibrations of cardiac valves as they open and close during the cardiac cycle. Turbulent blood flow may cause vibrations around the cardiac valves that create audible heart sounds and abnormal waveforms in the PCG, which are called *murmurs*. Different kinds of murmurs exist, and they are characterized in various ways, including location, timing, duration, shape, intensity, and pitch. The identification and analysis of murmurs can provide valuable information about cardiovascular pathologies.

However, while cardiac auscultation itself is relatively accessible, experts are needed to interpret heart sounds and PCGs, limiting the accessibility of auscultation for cardiac disease management, especially in resource-constrained environments, where limited access to clinical experts and infrastructure prevents widespread screening. The ability to correctly interpret PCGs for detecting murmurs and identifying different pathologies requires time and broad clinical experience. Therefore, the objective interpretation of the PCG remains a difficult skill to acquire.

The 2022 George B. Moody PhysioNet Challenge (formerly the PhysioNet/Computing in Cardiology Challenge) provided an opportunity to address these issues by inviting teams to develop automated approaches for detecting heart murmurs and abnormal cardiac function from PCG recordings. We sourced and shared PCGs from a largely pediatric population in Brazil, and we asked teams to use the PCGs to identify both heart murmurs and the clinical outcomes of a full diagnostic screening. The Challenge explored the potential of algorithmic approaches for interpreting heart sound recordings as part of diagnostic screening for more accessible cardiac care.

## Methods

### Challenge data

The CirCor DigiScope dataset [6] was used for the George B. Moody PhysioNet Challenge 2022. This dataset consists of 5268 PCG recordings from one or more auscultation locations during 1568 patient encounters with 1452 distinct patients. The patient population was primarily pediatric, ranging from neonates to adolescents, but it also included pregnant adults; no recordings were collected from fetuses. The dataset was collected during two screening campaigns in Paraíba, Brazil from July 2014 to August 2014 and from June 2015 to July 2015. The study protocol was approved by the 5192-Complexo Hospitalar HUOC/PROCAPE Institutional Review Board under the request of the Real Hospital Português de Beneficiência em Pernambuco. A detailed description of the dataset can be found in [6].

During the data collection sessions, each participant answered a sociodemographic questionnaire, followed by a clinical examination (anamnesis and physical examination), a nursing assessment (physiological measurements), and cardiac investigations (cardiac auscultation, chest radiography, electrocardiogram, and echocardiogram). The collected data were then analyzed by an expert pediatric cardiologist. The expert could re-auscultate the participant and/or request complementary tests. At the end of a session, the pediatric cardiologist either directed the participant for a follow-up appointment, referred the participant to cardiac catheterization or heart surgery, or discharged the participant.

The PCGs were recorded using an electronic auscultation device, the Littmann 3200 stethoscope, from up to four auscultation locations on the body; see Fig 1:

- Aortic valve: second intercostal space, right sternal border;

- Pulmonic valve: second intercostal space, left sternal border;

- Tricuspid valve: lower left sternal border; and

- Mitral valve: fifth intercostal space, midclavicular line (cardiac apex).

The choice of locations, the number of recordings at each location, and the duration of the PCGs varied between patients. The PCGs were recorded by multiple operators, but the PCGs for each patient encounter were recorded by a single operator, and they were recorded in a sequential manner, i.e., not simultaneously. The PCGs were also inspected for signal quality and semi-automatically segmented using the three algorithms proposed in [7],[8], and [9] and then corrected, as deemed necessary, by a cardiac physiologist.

Each patient's PCGs and clinical notes were also annotated for murmurs and abnormal cardiac function (described below). These annotations served as the labels for the Challenge.

The murmur annotations and characteristics (location, timing, shape, pitch, quality, and grade) were manually identified by a single cardiac physiologist independently of the available clinical notes and PCG segmentation. The cardiac physiologist annotated the PCGs by

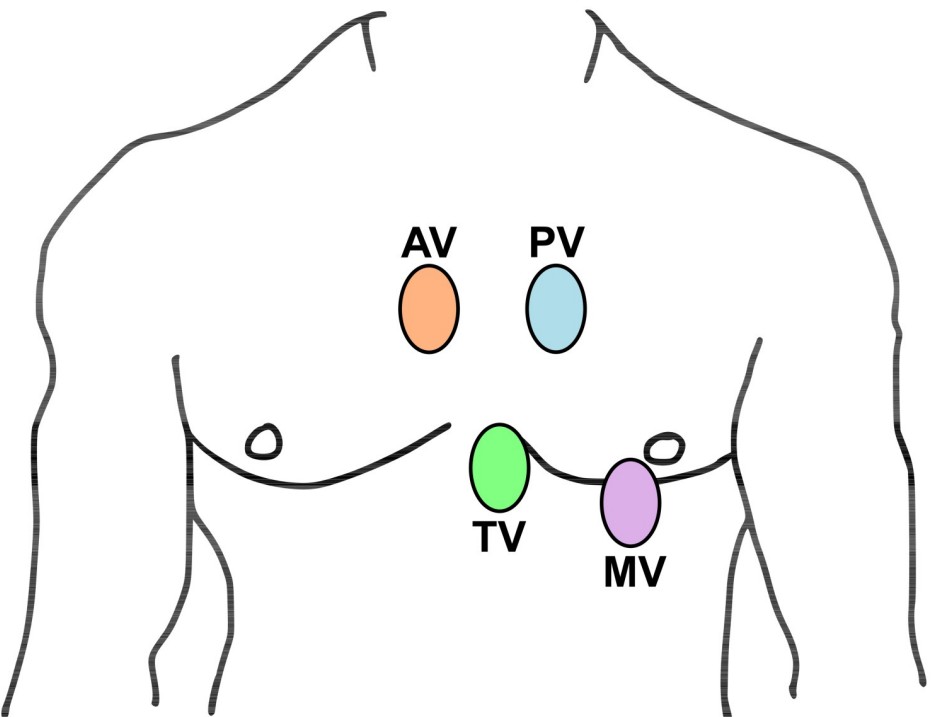

**Fig 1. Auscultation locations for the CirCor DigiScope dataset [6], which was used for the Challenge: Pulmonary valve (PV), aortic valve (AV), mitral valve (MV), and tricuspid valve (TV).**

listening to the audio recordings and by visually inspecting the corresponding waveforms. The murmur annotations indicated whether the annotator could detect the presence or absence of a murmur in a patient from the PCG recordings for the patient or whether the annotator was unsure about the presence or absence of a murmur. The murmur annotations did not indicate whether a murmur was pathological or innocent.

The clinical outcome annotations were determined by cardiac physiologists using all available clinical notes, including the socio-demographic questionnaire, clinical examination, nursing assessment, and cardiac investigations. In particular, these notes include reports from an echocardiogram, which is a standard diagnostic tool for characterizing cardiac function. The clinical outcome annotations indicated whether the expert annotator identified normal or abnormal cardiac function. The clinical outcome annotations were performed by different experts, and these experts were different from the expert who performed the murmur annotations.

In total, the Challenge dataset consisted of 5272 annotated PCG recordings from 1568 patient encounters with 1452 patients. We released 60% of the recordings in a public training set and retained 10% of the recordings in a private validation set and 30% of the recordings in a private test set. The training, validation, and test sets were matched to approximately preserve the univariate distributions of the variables in the data. Data from patients who participated in multiple screening campaigns belonged to only one of the training, validation, or test sets to prevent data leakage. We shared the training set at the beginning of the Challenge to allow the participants to develop their algorithms, and we sequestered the validation and test sets during the Challenge to evaluate the submitted the algorithms.

**Table 1. Demographic, murmur, and clinical outcome information in the Challenge training, validation, and/or test sets; `nan` values indicate unknown or missing values.**

| Variable | Description | Possible values | Dataset splits |
|---|---|---|---|
| Age | Reported age | Neonate (birth to 27 days), Infant (28 days to 1 year), Child (1 to 11 years), Adolescent (12 to 18 years), `nan` (unknown) | Training, validation, test |
| Sex | Reported sex | Female, Male | Training, validation, test |
| Height | Height in centimeters | Positive number or `nan` (unknown) | Training, validation, test |
| Weight | Weight in kilograms | Positive number or `nan` (unknown) | Training, validation, test |
| Pregnancy status | Reported pregnancy status | True, False | Training, validation, test |
| Murmur | Indicates if a murmur is present, absent, or unidentifiable or unknown for the annotator; a Challenge label | Present, Absent, Unknown | Training |
| Murmur locations | Auscultation locations for observed murmurs | PV, TV, AV, MV, Phc, `nan` (unknown); concatenated with + | Training |
| Most audible location | Auscultation location where murmurs sounded most intense | PV, TV, AV, MV, Phc, `nan` (unknown) | Training |
| Systolic murmur timing | Timing of the murmur within the systolic period | Early-systolic, Mid-systolic, Late-systolic, Holosystolic, `nan` (unknown) | Training |
| Systolic murmur shape | Shape of the murmur in the systolic period | Crescendo, Decrescendo, Diamond, Plateau, `nan` (unknown) | Training |
| Systolic murmur pitch | Pitch of the murmur in the systolic period | Low, Medium, High, `nan` (unknown) | Training |
| Systolic murmur grading | Grading of the murmur in the systolic period according to a modified Levine scale [10] | I/VI, II/VI, III/VI, `nan` (unknown) | Training |
| Systolic murmur quality | Quality of the murmur in the systolic period | Blowing, Harsh, Musical, `nan` (unknown) | Training |
| Diastolic murmur timing | Timing of the murmur within the diastolic period | Early-diastolic, Mid-diastolic, Holodiastolic, `nan` (unknown) | Training |
| Diastolic murmur shape | Shape of the murmur in the diastolic period | Decrescendo, Plateau, `nan` (unknown) | Training |
| Diastolic murmur pitch | Pitch of the murmur in the diastolic period | Low, Medium, High, `nan` (unknown) | Training |
| Diastolic murmur grading | Grading of the murmur in the diastolic period | I/IV, II/IV, III/IV, `nan` (unknown) | Training |
| Diastolic murmur quality | Quality of the murmur in the diastolic period | Blowing, Harsh, `nan` (unknown) | Training |
| Outcome | Indicates a normal or abnormal clinical outcome as diagnosed by the medical expert; a Challenge label | Normal, Abnormal | Training |
| Campaign | Screening campaign attended by the patient | CC2014, CC2015 | Training |
| Additional ID | Other patient identifier for patients who attended both screening campaigns | Patient identifier | Training |

Table 1 summarizes the variables provided in the training, validation, and test sets of the Challenge data. Table 2 summarizes the distributions of the variables provided with the data.

## Challenge objective

We designed the Challenge to explore the potential for algorithmic pre-screening of heart murmurs and abnormal heart function, especially in resource-constrained environments [18]. We asked the Challenge participants to design working, open-source algorithms for detecting heart murmurs and abnormal cardiac function from PCG recordings. For each patient encounter, each algorithm interpreted the PCG recordings and/or demographic data.

**Table 2. Demographic, murmur, and clinical outcome distributions across the Challenge training, validation, and test data.** For categorical variables, the entries of the table denote the fraction of the dataset with each possible value. For numerical variables, the entries of the table denote the median and first and third quartiles, respectively, of the values in the dataset, i.e., median [Q1, Q3].

| Variable | Training set | Validation set | Test set | Entire dataset |
|---|---|---|---|---|
| Age | | | | |
| - Neonate | 0.006 | 0.007 | 0.006 | 0.006 |
| - Infant | 0.134 | 0.101 | 0.107 | 0.122 |
| - Child | 0.705 | 0.678 | 0.723 | 0.708 |
| - Adolescent | 0.076 | 0.094 | 0.099 | 0.085 |
| - nan | 0.079 | 0.121 | 0.065 | 0.078 |
| Sex | | | | |
| - Female | 0.516 | 0.530 | 0.453 | 0.498 |
| - Male | 0.484 | 0.470 | 0.547 | 0.502 |
| Height | 115 [89, 133] cm | 116 [89, 136] cm | 113 [92, 135] cm | 115 [89, 134] cm |
| Weight | 20.4 [12.5, 31.2] kg | 20.9 [12.9, 34.6] kg | 21.0 [13.4, 32.7] kg | 20.6 [12.7, 32.0] kg |
| Pregnancy status | | | | |
| - False | 0.926 | 0.899 | 0.948 | 0.930 |
| - True | 0.074 | 0.101 | 0.052 | 0.070 |
| Murmur | | | | |
| - Absent | 0.738 | 0.711 | 0.719 | 0.730 |
| - Unknown | 0.072 | 0.107 | 0.073 | 0.076 |
| - Present | 0.190 | 0.181 | 0.208 | 0.195 |
| Outcome | | | | |
| - Normal | 0.516 | 0.617 | 0.493 | 0.518 |
| - Abnormal | 0.484 | 0.383 | 0.507 | 0.482 |
| Campaign | | | | |
| - CC2014 | 0.409 | 0.403 | 0.436 | 0.416 |
| - CC2015 | 0.591 | 0.597 | 0.564 | 0.584 |

**Challenge timeline.** The George B./ Moody PhysioNet Challenge 2022 was the 23rd George B. Moody PhysioNet Challenge [11, 18]. As with previous Challenges, the 2022 Challenge had an unofficial phase and an official phase.

The unofficial phase (February 1, 2022 to April 8, 2022) introduced the teams to the Challenge. We publicly shared the Challenge objective, training data, example algorithms, and evaluation metrics at the beginning of the unofficial phase. At this time, we only had access to the patients' murmur annotations, so we only asked the teams to detect murmurs. We invited the teams to submit the entries with the code for their algorithms for evaluation, and we scored at most 5 entries from each team on the hidden validation set during the unofficial phase.

Between the unofficial phase and official phase, we took a hiatus (April 9, 2022 to April 30, 2022) to improve the Challenge in response to feedback from the teams, the broader community, and our collaborators. During this time, we added the patients' clinical outcomes for abnormal cardiac function to the CirCor DigiScope dataset [6].

The official phase (May 1, 2022 to August 15, 2022) allowed the teams to refine their approaches for the Challenge. We updated the Challenge objectives, data, example algorithms, and evaluation metric at the beginning of the official phase. At this time, we had access to both the patients' murmur annotation and clinical outcomes, so we asked the teams to detect murmurs and abnormal cardiac function. We again invited the teams to submit their entries for evaluation, and we scored at most 10 entries from each team on the hidden validation set during the official phase.

After the end of the official phase, we asked each team to choose a single entry from their team for evaluation on the test set. We allowed the teams to choose any successful model from the official phase, but most teams chose their best-scoring entries. We only evaluated one entry from each team to prevent sequential training on the test set. The winners of the Challenge were the teams with the best scores on the test set. We announced the winners at the end of the Computing in Cardiology (CinC) 2022 conference.

The teams presented and defended their work at CinC 2022, which was held in Tampere, Finland. As described in the next section, the teams wrote four-page conference proceeding paper describing their work, which we reviewed for accuracy and coherence. The code for their algorithms will be publicly released after the end of the Challenge and the publication of the papers on the Challenge website.

**Challenge rules and expectations.** We encouraged the teams to ask questions, pose concerns, and discuss the Challenge in a public forum, but we prohibited them from discussing or sharing their work during the unofficial phase, hiatus, or official phase of the Challenge to preserve the diversity and uniqueness of their approaches.

For both phases of the Challenge, we required the teams to submit the complete code for their algorithms, including their preprocessing, training, and inference steps. We first ran each team's training code on the public training set to train the models. We then ran the trained models on the hidden validation and test sets to label the data; we ran the trained models on each patient sequentially to reflect the sequential nature of the screening process. We then scored the outputs from the models using the expert annotations on hidden validation and test sets.

We allowed the teams to submit either MATLAB or Python code; other programming languages were considered upon request, but there were no requests for other programming languages. Participants containerized their code in Docker and submitted it by sharing private GitHub or Gitlab repositories with their code. We downloaded their code and ran it in containerized environments on Google Cloud. We described the computational architecture of these environments more fully in [12].

Each entry had access to 8 virtual CPUs, 52 GB RAM, 50 GB local storage, and an optional NVIDIA T4 Tensor Core GPU (driver version 470.82.01) with 16 GB VRAM. We imposed a 72-hour time limit for training each entry on the training set without a GPU, a 48-hour time limit for training each entry on the training set with a GPU, and a 24-hour time limit for running each trained entry on either the validation or test set either with or without a GPU.

To aid the teams, we shared example MATLAB and Python entries. These examples used random forest classifiers with the age group, sex, height, weight, pregnancy status of the patient as well as the presence, mean, variance, and skewness of the numerical values in each PCG recording as features. We did not design these example entries to perform well. Instead, we designed them to provide minimal working examples of how to read the Challenge data and write the model outputs.

**Challenge evaluation.** To capture the focus of the 2022 Challenge on algorithmic screening for heart murmurs and abnormal cardiac function, we developed novel scoring metrics for each of the two Challenge tasks: detecting heart murmurs and identifying clinical outcomes for abnormal or normal heart function.

As described above, the murmurs were directly observable from the PCG recordings, but the clinical outcomes were determined by a more comprehensive diagnostic screening, including the interpretation of an echocardiogram. However, despite these differences, we asked the teams to perform both tasks using only the PCGs and routine demographic data, which allowed us to explore the diagnostic potential of algorithmic approaches for the interpretation of relatively easily accessible PCGs.

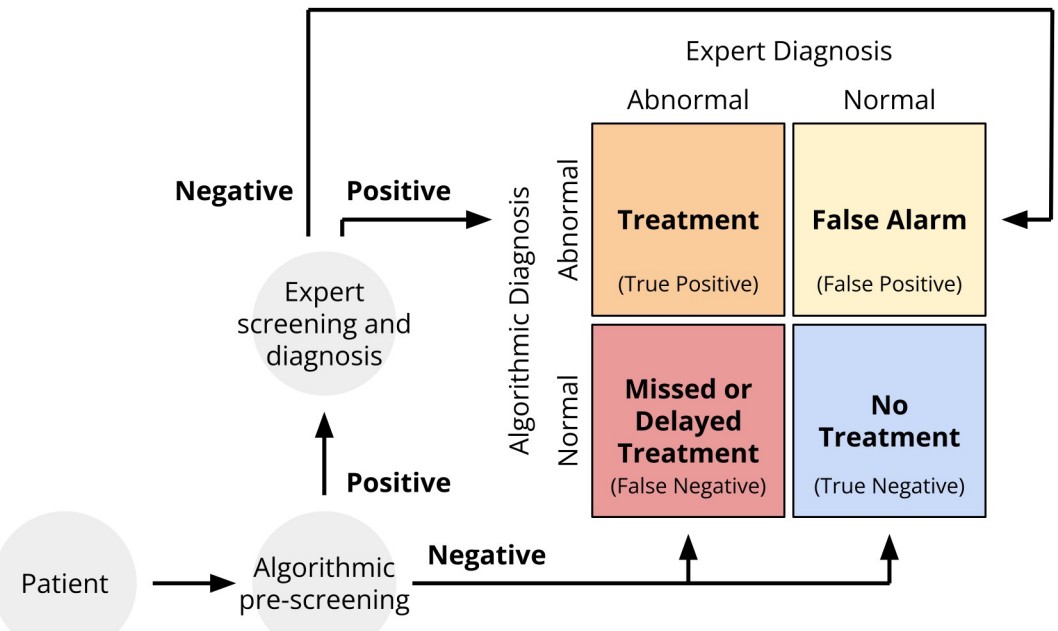

**Fig 2. Screening and diagnosis pipeline for the Challenge.** All patients would receive algorithmic pre-screening, and patients with positive results from algorithmic pre-screening would receive confirmatory expert screening and diagnosis. (i) Patients with positive results from algorithmic pre-screening and expert annotators would receive treatment; they are true positive cases. Patients with positive results from algorithmic pre-screening and negative results from expert annotators would not receive treatment; they are false positive cases or false alarms. Patients with negative results from algorithmic pre-screening who would have received positive results from the expert annotators would have missed or delayed treatment; they are false negative cases. Patients with negative results from algorithmic pre-screening who would have also received negative results from expert annotators also would not receive treatment; they are true negative cases.

The algorithms for both of these tasks effectively pre-screened patients for expert referral. Under this paradigm, if an algorithm inferred potentially abnormal cardiac function, i.e., the model outputs were murmur present, murmur unknown, or outcome abnormal, then the algorithm would refer the patient to a human expert for a confirmatory diagnosis and potential treatment. If the algorithm inferred normal cardiac function, i.e., if the model outputs were murmur absent or outcome normal, then the algorithm would not refer the patient to an expert and the patient would not receive treatment. Fig 2 illustrates this algorithmic pre-screening process as part of a larger diagnostic pipeline.

For the murmur detection task, we introduced a weighted accuracy metric that assessed the ability of an algorithm to reproduce the results of a skilled human annotator. For the clinical outcome identification task, we introduced a cost-based scoring metric that reflected the cost of expert diagnostic screening as well as the costs of timely, missed, and delayed treatment for abnormal cardiac function. The team with the highest weighted accuracy metric won the murmur detection task, and the team with the lowest cost-based evaluation metric won the clinical outcome identification task.

We formulated versions of both of these evaluation metrics for both tasks to allow for more direct comparisons; see S1 Appendix for the additional metrics. We also calculated several traditional evaluation metrics to provide additional context to the performance of the models.

Cost-based scoring is controversial, in part, because healthcare costs are an imperfect proxy for health needs [13, 14]; we reflect on this important issue in the Discussion section. However, screening costs necessarily limit the ability to perform screening, especially in more resource-

constrained environments, so we considered costs as an imperfect proxy for improving access to cardiac screening.

**Weighted accuracy metric.** We introduced a weighted accuracy metric to evaluate the murmur detection algorithms. This metric assessed the ability of the algorithms to reproduce the decisions of the expert annotator. This weighted accuracy metric is similar to the traditional accuracy metric, but it assigned more weight to patients that had or potentially had murmurs than to patients that did not have murmurs. These weights reflect the rationale that, in general, a missed diagnosis is more harmful than a false alarm.

We defined the weighted accuracy metric for the murmur detection task as

$$a_{\mathrm{murmur}} = \frac{5m_{\mathrm{PP}} + 3m_{\mathrm{UU}} + m_{\mathrm{AA}}}{5(m_{\mathrm{PP}} + m_{\mathrm{UP}} + m_{\mathrm{AP}}) + 3(m_{\mathrm{PU}} + m_{\mathrm{UU}} + m_{\mathrm{AU}}) + (m_{\mathrm{PA}} + m_{\mathrm{UA}} + m_{\mathrm{AA}})}, \quad (1)$$

where Table 3 defines a three-by-three confusion matrix $M = [m_{ij}]$ for the murmur present, murmur unknown, and murmur absent classes.

These coefficients were chosen to reflect the trade-off between false positives and false negatives, where clinicians may tolerate multiple false alarms to avoid a single missed diagnosis. In (1), murmur present cases have five times the weight of murmur absent cases, and the murmur unknown cases have three times the weight of murmur absent cases, to reflect a tolerance of five false alarms for every one false positive.

Like the traditional accuracy metric, this metric only rewarded algorithms for correctly labeling patients, but it provided the highest reward for correctly classifying patients with murmurs and the lowest reward for correctly classifying patients without murmurs. It provided an intermediate reward for correctly classifying patients of unknown murmur status to reflect the difficulty and importance of indicating when the recordings are not adequate for diagnosis.

We used (1) to rank the Challenge algorithms for the murmur detection task. The team with the highest value of (1) won this task.

**Cost-based evaluation metric.** We introduced a cost-based evaluation metric to evaluate the clinical outcome algorithms for abnormal or normal heart function. This metric considered the ability of the algorithms to reduce the costs associated with diagnosing and treating patients, primarily by having experts screen fewer patients with normal cardiac function. We again emphasize that healthcare costs are an imperfect surrogate for health needs [13, 14]. However, costs remain a practical consideration for resource-constrained environments.

For each patient encounter, the algorithm interpreted the PCG recordings and demographic data for the encounter. Under this paradigm, if an algorithm inferred abnormal cardiac function, then it would refer the patient to a human expert for a confirmatory diagnosis. If the expert confirmed the diagnosis, then the patient would receive treatment, and if the expert did not confirm the diagnosis, then the patient would not receive treatment. Alternatively, if the algorithm inferred normal cardiac function, then it would not refer the patient to

**Table 3. Confusion matrix *M* for murmur detection with three classes: Murmur present, murmur unknown, and murmur absent.** The columns are the ground truth labels from the human annotator, and the rows are the model outputs. The entries of the confusion matrix provide the number of patients with each model output for each ground truth class.

| | | Murmur Expert | | |
| --- | --- | --- | --- | --- |
| | | **Present** | **Unknown** | **Absent** |
| Murmur Classifier | Present | $m_{\mathrm{PP}}$ | $m_{\mathrm{PU}}$ | $m_{\mathrm{PA}}$ |
| | Unknown | $m_{\mathrm{UP}}$ | $m_{\mathrm{UU}}$ | $m_{\mathrm{UA}}$ |
| | Absent | $m_{\mathrm{AP}}$ | $m_{\mathrm{AU}}$ | $m_{\mathrm{AA}}$ |

an expert, and the patient would not receive treatment, even if the patient had abnormal cardiac function that would have been detected by a human expert.

We associated each of these steps with a cost: the cost of algorithmic pre-screening, the cost of expert screening, the cost of timely treatment, and the cost of delayed or missed treatment.

For simplicity, we assumed that algorithmic pre-screening had a relatively small cost that depended linearly on the number of algorithmic pre-screenings. We also assumed that both timely treatments and delayed or missed treatments had relatively large costs that, on average, depended linearly on the number of patients with delayed or missed treatment. Given our focus on the ability of algorithmic pre-screening to reduce human screening of patients with normal cardiac function, we assumed that expert screening had an intermediate cost that depended non-linearly on the number of screenings as well on as the infrastructure and capacity of the healthcare system. Of course, treatment costs are also non-linear in the number of treated patients for similar reasons, but non-urgent treatments can arguably utilize the capacity of the broader healthcare system. Screening far below the capacity of the healthcare system was inefficient and incurred a low total cost but high average cost. Screening above the capacity of the healthcare system was also inefficient and incurred both a high average cost and a high total cost.

Therefore, we introduced the following cost-based evaluation metric for identifying clinical outcomes. We defined the total cost of diagnosis and treatment with algorithmic pre-screening as

$$
\begin{aligned}
c_{\text{outcome}}^{\text{total}} = \ & f_{\text{algorithm}}(n_{\text{patients}}) \\
& + f_{\text{expert}}(n_{\text{TP}} + n_{\text{FP}}, \ n_{\text{patients}}) \\
& + f_{\text{treatment}}(n_{\text{TP}}) \\
& + f_{\text{error}}(n_{\text{FN}}),
\end{aligned}
\tag{2}
$$

where Table 4 defines a two-by-two confusion matrix $N = [n_{ij}]$ for the classes for abnormal and normal cardiac function, $n_{\text{patients}} = n_{\text{TP}} + n_{\text{FP}} + n_{\text{FN}} + n_{\text{TN}}$ is the total number of patient encounters, and $f_{\text{algorithm}}, f_{\text{expert}}, f_{\text{treatment}}, f_{\text{error}}$ are defined below.

Again, for simplicity, we assumed that the costs for algorithmic pre-screening, timely treatments, and missed or late treatments were linear. We defined

$$
f_{\text{algorithm}}(s) = 10s
\tag{3}
$$

as the total cost of $s$ pre-screenings by an algorithm,

$$
f_{\text{treatment}}(s) = 10000s
\tag{4}
$$

**Table 4. Confusion matrix $N$ for clinical outcome detection with two classes: Clinical outcome abnormal and clinical outcome normal.** The columns are the ground truth labels from the human annotator, and the rows are the classifier outputs. The entries of the confusion matrix provide the number of patients with each classifier output for each ground truth class.

| | | Clinical Outcome Expert | |
|---|---|---|---|
| | | **Abnormal** | **Normal** |
| Clinical Outcome Classifier | Abnormal | $n_{\text{TP}}$ | $n_{\text{FP}}$ |
| | Normal | $n_{\text{FN}}$ | $n_{\text{TN}}$ |

as the total cost of $s$ treatments, and

$$f_{\text{error}}(s) = 50000s \tag{5}$$

as the total cost of $s$ missed or delayed treatments.

Conversely, to better capture the potential benefits of algorithmic pre-screening, we assumed that the cost for expert screening was non-linear. We defined

$$f_{\text{expert}}(s, t) = \left(25 + 397\frac{s}{t} - 1718\frac{s^2}{t^2} + 11296\frac{s^4}{t^4}\right)t \tag{6}$$

as the total cost of $s$ screenings by a human expert out of a population of $t$ patient encounters so that

$$g_{\text{expert}}(x) = 25 + 397x - 1718x^2 + 11296x^4 \tag{7}$$

was the mean cost of screening a fraction $x = s/t$ of the patient cohort by an expert; this reparameterization of (6) allowed us to compare algorithms on datasets with different numbers of patients. We designed (6) and (7) so that the mean cost of expert screening was lowest when only 25% of the patient cohort received an expert screening to reflect the optimal capacity of the screening system. Figs 3 and 4 show these costs across different patient cohort and screening sizes, and S2 Appendix provides a fuller derivation of (6) and (7).

To more easily compare costs across databases with different numbers of patients, e.g., the training, validation, and test sets, we defined the mean per-patient cost of diagnosis and treatment with algorithmic pre-screening as

$$c_{\text{outcome}} = \frac{c_{\text{outcome}}^{\text{total}}}{n_{\text{patients}}}. \tag{8}$$

We used (8) to rank the Challenge algorithms for the murmur detection task. The team with the lowest value of (8) won this task.

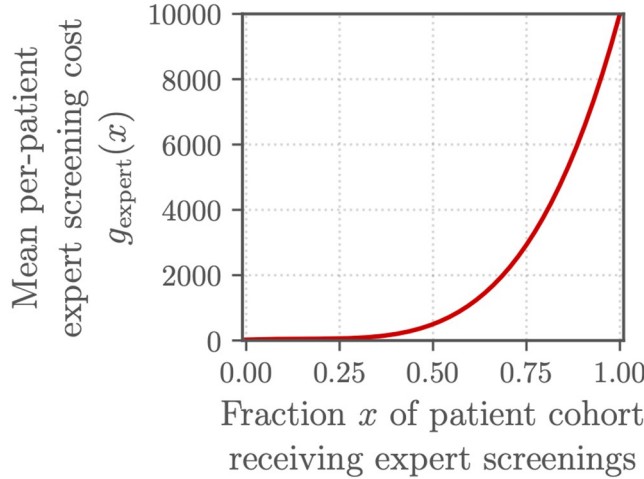

**Fig 3. The expert screening cost $g_{\text{expert}}(x)$ defined for the Challenge: Mean cost for screening a fraction $x$ of a patient cohort for cardiac abnormalities.** Mean **per-patient** expert screening cost $g_{\text{expert}}(x)$, i.e., the total expert screening cost for a patient cohort divided by the number of patients in the cohort.

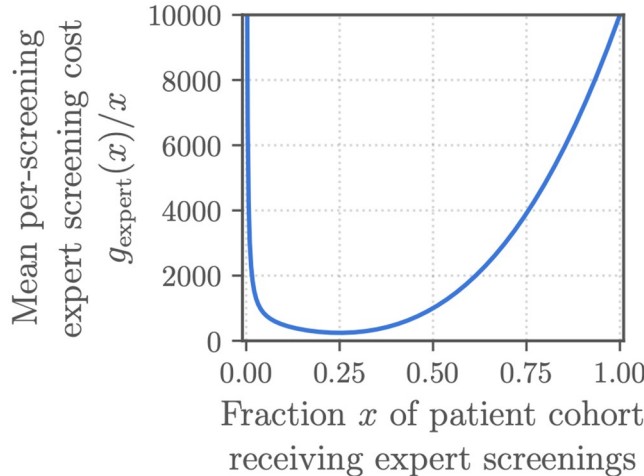

**Fig 4. The expert screening cost $g_{expert}(x)$ defined for the Challenge: Mean cost for screening a fraction $x$ of a patient cohort for cardiac abnormalities.** Mean **per-screening** expert screening cost $g_{expert}(x)/x$, i.e., the total expert screening cost for a patient cohort divided by the number of patients in the cohort and the fraction of expert screenings for the cohort.

## Challenge results

### Challenge entries

We received 779 entries from 87 teams throughout the course of the 2022 PhysioNet Challenge, resulting in 77 submitted CinC abstracts, 62 accepted CinC abstracts, 43 published CinC proceedings papers, and 53 final entries with working code from 53 different teams.

These entries represent a diversity of approaches to the Challenge. A total of 81 teams submitted 293 entries during the unofficial phase of the Challenge, and a total of 63 teams submitted 486 entries during the official phase of the Challenge. Of the 779 total entries, we received 652 entries from 75 teams that were implemented in Python, and 127 entries from 17 teams that were implemented in MATLAB. We received 519 entries from 60 teams that requested a graphics processing unit (GPU) for their entries, and 260 entries from 49 teams requested only a central processing unit (CPU), i.e., no GPU. In total, we received 473 successful entries that we were able to train on the public training set and evaluate on the hidden validation set, and 306 entries that we were unable to train on the training set and/or evaluate on the validation set due to various errors in the submitted code.

We allowed each team with a successful entry during the official phase to select a single entry for evaluation on the test set. A total of 58 teams had a successful entry during the official phase that we were able to train on the training set and evaluate on the validation set. Most of these teams chose their best-scoring entry from the official phase, but some teams selected their most recent entry or another entry, often because they thought that another entry with a lower score on the validation set would better generalize to the test set.

In some cases, a team's best-scoring entry for the murmur detection task was different from the team's best-scoring entry for the clinical outcome identification task because of the differences between the tasks and the evaluation metrics for grading them; in these cases, we chose a different entry for each task. However, since the teams could implement different approaches for each task, and since they could have submitted these different approaches as part of the same entry, we did not distinguish between teams and entries that submitted their best scoring

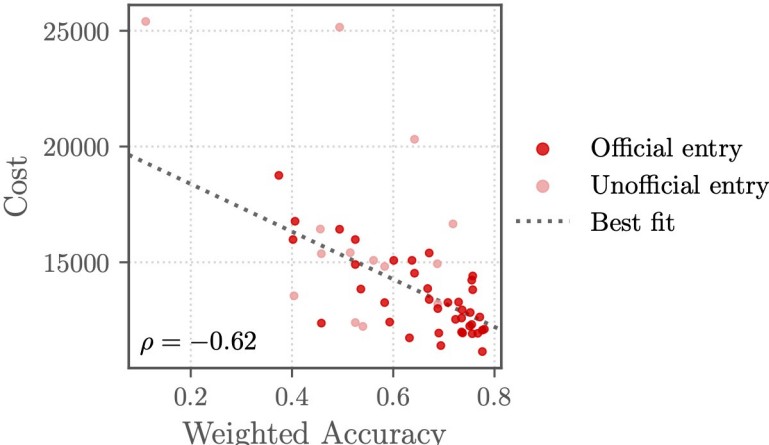

**Fig 5. Weighted accuracy metric (1) for the murmur detection task (*x*-axis) and the cost metric (8) scores for the clinical outcome identification task (*y*-axis) of the final Challenge entries on the hidden test set.** Each point shows an entry, and the shading of each point shows whether the entry was an official entry (dark red points) that satisfied all of the Challenge rules or an unofficial entry (light red points) that did not. The Spearman correlation coefficient $\rho$ between the scores is given in the plot, and a line of best fit (gray dotted line) is given by a robust linear regression.

entries in the same code submission or in different code submissions. We will share the best scoring entries for both whether they were part of the same or different code submissions.

Of the 58 teams with a successful entry during the official phase, 53 teams had code that we were able to score on the training, validation, and test sets for both the murmur detection and clinical outcome identification tasks, resulting in 53 working entries to the Challenge.

Fig 5 compares the performance of the working Challenge entries on the murmur detection and clinical outcome identification tasks. Figs 6 and 7 compare the Challenge weighted accuracy and cost metrics, respectively, with traditional scoring metrics, including area under the

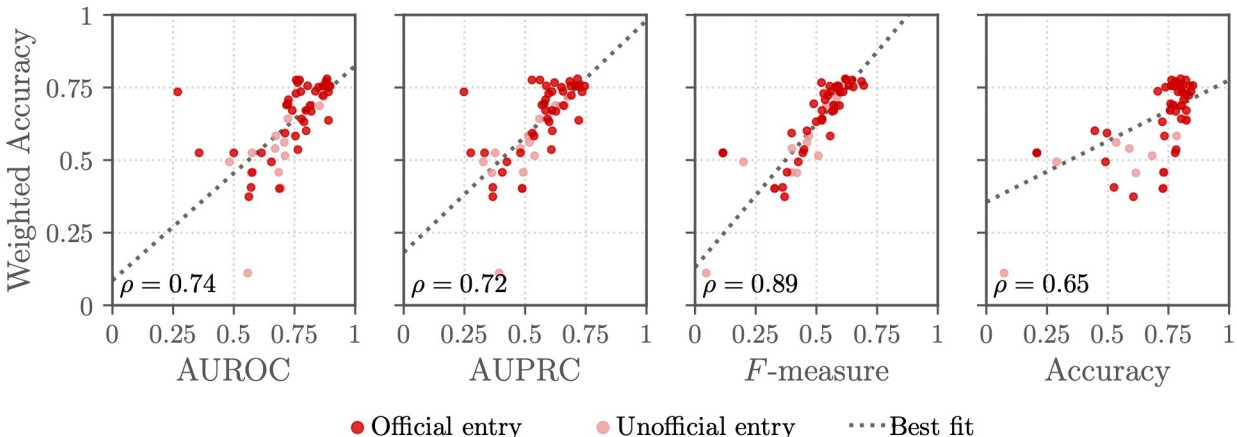

**Fig 6. Traditional evaluation metric (*x*-axis) and weighted accuracy metric (*y*-axis) scores for the final Challenge entries with the murmur detection task on the hidden test set.** AUROC is the area under the receiver operating character curve, AUPRC is the area under the precision-recall curve, *F*-measure is the macro-averaged *F*-measure, and accuracy is the traditional accuracy metric. Each point shows an entry, and the shading of each point shows whether the entry was an official entry (dark red points) that satisfied all of the Challenge rules or an unofficial entry (light red points) that did not. The Spearman correlation coefficients $\rho$ between the metrics are given in the individual plots, and a line of best fit (gray dotted line) is given by a robust linear regression.

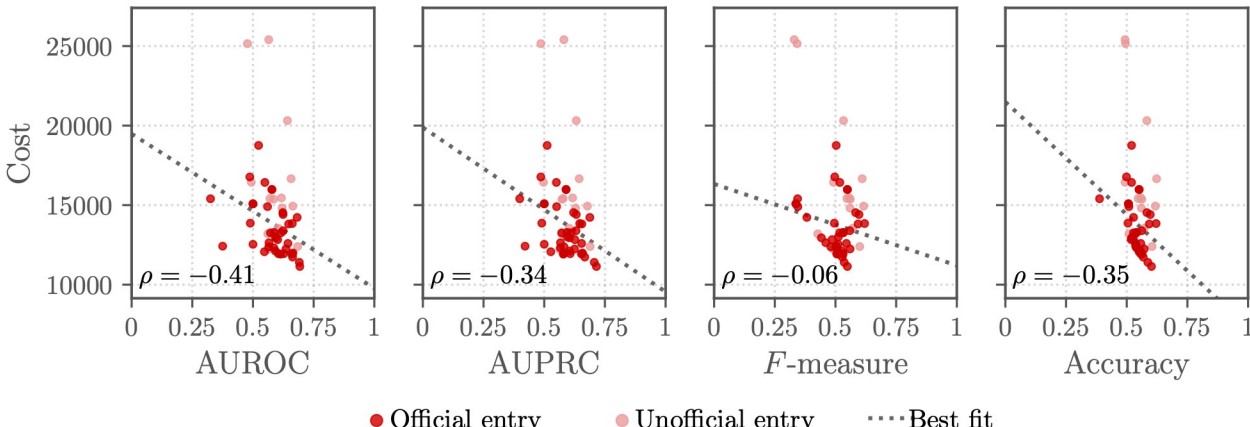

**Fig 7. Traditional evaluation metric (*x*-axis) and cost metric (*y*-axis) scores for the final Challenge entries with the clinical outcome identification task on the hidden test set.** AUROC is the area under the receiver operating character curve, AUPRC the is area under the precision-recall curve, *F*-measure is the traditional *F*-measure, and accuracy is the traditional accuracy metric. Each point shows an entry, and the shading of each point shows whether the entry was an official entry (dark red points) that satisfied all of the Challenge rules or an unofficial entry (light red points) that did not. The Spearman correlation coefficients $\rho$ between the metrics are given in the individual plots, and a line of best fit (gray dotted line) is given by a robust linear regression.

receiver operating characteristic curve (AUROC), the area under the precision-recall curve (AUPRC), the macro-averaged *F*-measure, and the traditional accuracy metric.

For the official rankings, we required the teams to have successful unofficial and official phase entries, to have an accepted CinC abstract, to publicly share their CinC proceedings paper preprint by the CinC preprint submission deadline, to have an accepted final CinC proceedings paper by the CinC final paper submission deadline, and to license their code under an open-source or similarly permissive license. We also allowed teams without a successful unofficial phase entry and/or accepted CinC abstract who had a successful, high-scoring official phase entry to submit a CinC abstract after the original abstract submission deadline as a "wild card" team. The wild card abstract and proceedings paper were otherwise subject to the same review process as for the other participants.

We imposed these requirements to support the dissemination of research into the research community. Of the 53 working entries, a total of 40 teams were officially ranked. Tables 5 and 6 summarize the traditional and Challenge evaluation metrics for the officially ranked entries for the murmur detection and clinical outcome identification tasks, respectively, on the hidden test set.

To assess the robustness of the algorithms, we tested whether they actually learned from the training set; algorithms that cannot learn from new data have more limited use. To do so, we trained them on a subset of the training set with permuted labels and ran the retrained models on the validation and test sets with the original labels; we did not change any of the publicly available training data or labels but simply the size of the training set and the relationship between the data and the labels in the training set. Algorithms that truly learned from the training set performed much worse when retrained on the modified training set, but algorithms that did not learn from the training set performed the same or better on the modified training set. Only 35 of the 53 working entries were robust enough for us to complete this process. The remaining 18 working entries either crashed or achieved the same or a better score when retrained on the modified training set, suggesting that they were insensitive to the training set labels. While we encouraged teams to submit robust code, we did not share the details

**Table 5. Scores of the officially ranked methods on the test set for the murmur detection task.** AUROC is the area under the receiver operating characteristic curve, AUPRC is the area under the precision-recall curve, *F*-measure is the macro-averaged *F*-measure, accuracy is the traditional accuracy metric, and weighted accuracy is the weighted accuracy metric (1), i.e., the metric used to rank algorithms for the murmur detection task.

| Rank | Team | AUROC | AUPRC | *F*-measure | Accuracy | Weighted Accuracy |
|---|---|---|---|---|---|---|
| * | *Voting—GBT* | *0.677* | *0.465* | *0.660* | *0.845* | *0.790* |
| ** | *Voting—RF* | *0.775* | *0.589* | *0.650* | *0.832* | *0.789* |
| 1 | HearHeart [19] | 0.884 | 0.716 | 0.619 | 0.801 | 0.780 |
| 2 | CUED_Acoustics [20] | 0.757 | 0.528 | 0.623 | 0.763 | 0.776 |
| 2 | HearTech+ [21] | 0.771 | 0.561 | 0.647 | 0.822 | 0.776 |
| 4 | PathToMyHeart [22] | 0.880 | 0.684 | 0.686 | 0.778 | 0.771 |
| 5 | CAU_UMN [23] | 0.763 | 0.621 | 0.521 | 0.786 | 0.767 |
| 6 | Care4MyHeart [24] | 0.891 | 0.717 | 0.695 | 0.851 | 0.757 |
| 6 | SmartBeatIT [25] | 0.873 | 0.707 | 0.633 | 0.809 | 0.757 |
| 8 | CeZIS [26] | 0.804 | 0.587 | 0.586 | 0.761 | 0.756 |
| 9 | ISIBrno-AIMT [27] | 0.897 | 0.746 | 0.555 | 0.839 | 0.755 |
| 9 | Murmur Mia! [28] | 0.867 | 0.688 | 0.592 | 0.771 | 0.755 |
| 11 | PhysioDreamfly [29] | 0.885 | 0.723 | 0.652 | 0.799 | 0.752 |
| 12 | Heart2Beat [30] | 0.849 | 0.650 | 0.579 | 0.738 | 0.751 |
| 13 | Listen2YourHeart [31] | 0.836 | 0.656 | 0.597 | 0.706 | 0.737 |
| 14 | Revenger [32] | 0.776 | 0.611 | 0.559 | 0.843 | 0.736 |
| 15 | matLisboa [33] | 0.269 | 0.248 | 0.610 | 0.807 | 0.735 |
| 15 | uke-cardio [34] | 0.890 | 0.735 | 0.597 | 0.790 | 0.735 |
| 17 | One_Heart_Health [35] | 0.756 | 0.597 | 0.530 | 0.820 | 0.729 |
| 18 | MetaHeart [36] | 0.868 | 0.691 | 0.548 | 0.834 | 0.723 |
| 19 | Team_IIITH [37] | 0.725 | 0.584 | 0.536 | 0.813 | 0.708 |
| 20 | prna [38] | 0.723 | 0.579 | 0.489 | 0.759 | 0.694 |
| 21 | HCCL [39] | 0.717 | 0.569 | 0.571 | 0.769 | 0.690 |
| 22 | amc-sh [40] | 0.814 | 0.658 | 0.595 | 0.803 | 0.688 |
| 23 | JUST_IT_Academy1 [41] | 0.797 | 0.610 | 0.572 | 0.757 | 0.671 |
| 23 | LSMU [42] | 0.741 | 0.584 | 0.523 | 0.824 | 0.671 |
| 25 | listNto_urHeart [43] | 0.819 | 0.627 | 0.565 | 0.780 | 0.668 |
| 26 | USST_Med [44] | 0.779 | 0.593 | 0.524 | 0.803 | 0.642 |
| 27 | PCGPAW [45] | 0.890 | 0.721 | 0.522 | 0.824 | 0.637 |
| 28 | Melbourne_Kangas [46] | 0.789 | 0.599 | 0.499 | 0.725 | 0.632 |
| 29 | SeaCrying [47] | 0.798 | 0.612 | 0.461 | 0.447 | 0.601 |
| 30 | Simulab [48] | 0.711 | 0.530 | 0.398 | 0.497 | 0.593 |
| 31 | MainLab [49] | 0.754 | 0.537 | 0.557 | 0.734 | 0.583 |
| 32 | Leicester Fox [50] | 0.764 | 0.608 | 0.449 | 0.782 | 0.536 |
| 33 | Eagles [51] | 0.357 | 0.276 | 0.115 | 0.208 | 0.525 |
| 33 | fly_h [52] | 0.614 | 0.481 | 0.444 | 0.778 | 0.525 |
| 33 | lubdub [53] | 0.500 | 0.333 | 0.115 | 0.208 | 0.525 |
| 36 | AKSJ_97BSc [54] | 0.655 | 0.425 | 0.425 | 0.491 | 0.494 |
| 37 | UKJ_FSU [55] | 0.576 | 0.406 | 0.379 | 0.730 | 0.458 |
| 38 | BrownBAI [56] | 0.571 | 0.367 | 0.361 | 0.526 | 0.406 |
| 39 | Bear_FH [57] | 0.688 | 0.488 | 0.328 | 0.727 | 0.402 |
| 40 | MobiHealth [58] | 0.562 | 0.367 | 0.369 | 0.606 | 0.374 |

**Table 6. Scores of the officially ranked methods on the test set for the clinical outcome identification task.** AUROC is the area under the receiver operating characteristic curve, AUPRC is the area under the precision-recall curve, *F*-measure is the traditional *F*-measure, accuracy is the traditional accuracy metric, and cost is the cost-based evaluation metric (8), i.e., the metric used to rank algorithms for the clinical outcome identification task.

| Rank | Team | AUROC | AUPRC | *F*-measure | Accuracy | Cost |
|---|---|---|---|---|---|---|
| 1 | CUED_Acoustics [20] | 0.693 | 0.715 | 0.549 | 0.602 | 11144 |
| * | *Voting-GBT* | *0.583* | *0.553* | *0.546* | *0.587* | *11357* |
| 2 | prna [38] | 0.691 | 0.706 | 0.536 | 0.587 | 11403 |
| ** | *Voting-RF* | *0.625* | *0.584* | *0.596* | *0.606* | *11687* |
| 3 | Melbourne_Kangas [46] | 0.663 | 0.667 | 0.528 | 0.568 | 11735 |
| 4 | CeZIS [26] | 0.614 | 0.579 | 0.511 | 0.560 | 11916 |
| 5 | CAU_UMN [23] | 0.660 | 0.653 | 0.505 | 0.562 | 11933 |
| 6 | HCCL [39] | 0.605 | 0.579 | 0.532 | 0.562 | 11943 |
| 7 | Listen2YourHeart [31] | 0.627 | 0.607 | 0.512 | 0.558 | 11946 |
| 8 | uke-cardio [34] | 0.665 | 0.658 | 0.509 | 0.556 | 11990 |
| 9 | HearTech+ [21] | 0.547 | 0.527 | 0.508 | 0.551 | 12069 |
| 10 | HearHeart [19] | 0.593 | 0.605 | 0.528 | 0.556 | 12110 |
| 11 | Heart2Beat [30] | 0.636 | 0.625 | 0.560 | 0.572 | 12244 |
| 12 | ISIBrno-AIMT [27] | 0.586 | 0.567 | 0.505 | 0.541 | 12313 |
| 13 | UKJ_FSU [55] | 0.565 | 0.582 | 0.481 | 0.539 | 12373 |
| 14 | Simulab [48] | 0.375 | 0.421 | 0.503 | 0.537 | 12419 |
| 15 | MetaHeart [36] | 0.500 | 0.500 | 0.511 | 0.537 | 12536 |
| 16 | matLisboa [33] | 0.644 | 0.654 | 0.545 | 0.558 | 12593 |
| 17 | PathToMyHeart [22] | 0.566 | 0.567 | 0.460 | 0.535 | 12637 |
| 18 | PhysioDreamfly [29] | 0.602 | 0.626 | 0.486 | 0.518 | 12831 |
| 19 | Revenger [32] | 0.594 | 0.604 | 0.442 | 0.520 | 12944 |
| 20 | amc-sh [40] | 0.593 | 0.594 | 0.512 | 0.528 | 13002 |
| 21 | MainLab [49] | 0.572 | 0.572 | 0.517 | 0.528 | 13259 |
| 22 | Team_IIITH [37] | 0.616 | 0.610 | 0.533 | 0.541 | 13264 |
| 23 | One_Heart_Health [35] | 0.587 | 0.594 | 0.531 | 0.539 | 13283 |
| 24 | JUST_IT_Academy1 [41] | 0.624 | 0.631 | 0.559 | 0.562 | 13394 |
| 25 | SmartBeatIT [25] | 0.647 | 0.655 | 0.593 | 0.593 | 13815 |
| 26 | Leicester Fox [50] | 0.662 | 0.646 | 0.620 | 0.621 | 13844 |
| 27 | listNto_urHeart [43] | 0.489 | 0.490 | 0.495 | 0.505 | 13866 |
| 28 | Murmur Mia! [28] | 0.682 | 0.688 | 0.382 | 0.528 | 14228 |
| 29 | Care4MyHeart [24] | 0.624 | 0.632 | 0.597 | 0.597 | 14410 |
| 30 | USST_Med [44] | 0.623 | 0.621 | 0.583 | 0.583 | 14529 |
| 31 | lubdub [53] | 0.560 | 0.551 | 0.345 | 0.509 | 14905 |
| 32 | PCGPAW [45] | 0.500 | 0.500 | 0.337 | 0.507 | 15083 |
| 32 | SeaCrying [47] | 0.500 | 0.500 | 0.337 | 0.507 | 15083 |
| 34 | LSMU [42] | 0.325 | 0.399 | 0.344 | 0.388 | 15402 |
| 35 | Bear_FH [57] | 0.578 | 0.590 | 0.549 | 0.551 | 15982 |
| 35 | fly_h [52] | 0.578 | 0.590 | 0.549 | 0.551 | 15982 |
| 37 | AKSJ_97BSc [54] | 0.549 | 0.550 | 0.518 | 0.520 | 16427 |
| 38 | BrownBAI [56] | 0.487 | 0.486 | 0.497 | 0.499 | 16773 |
| 39 | MobiHealth [58] | 0.523 | 0.512 | 0.503 | 0.520 | 18754 |

about how we would assess the robustness of their algorithms to encourage them to make their code robust in as many ways as possible. Therefore, since we did not provide advance notice of the exact requirements for robustness, we did not disqualify teams whose algorithms failed this test.

## Voting algorithms

While some entries had better overall performance than others, an entry with a lower overall performance may perform better on certain patient subgroups than another entry with a stronger overall performance. Therefore, an algorithm that can learn the different strengths of different algorithms can improve on the overall performance of the individual entries. To test this concept, we developed multiple voting algorithms to attempt to leverage the diversity of Challenge entries.

For the voting algorithms, we considered the discrete model outputs (see Tables 3 and 4) from the 39 algorithms from teams that were officially ranked for both the murmur detection and clinical outcome tasks (see Tables 5 and 6). We considered these tasks separately, and we used the relevant Challenge scoring metric to evaluate the voting algorithms for each task: the weighted accuracy metric (1) for the murmur detection task and the cost metric (8) for the clinical outcome identification task. We used the common gradient-boosting trees (GBT) and random forests (RF) models for the voting algorithms [15, 16, 17].

We developed and evaluated the GBT and RF voting algorithms using the following procedure. First, we trained the GBT and RF voting algorithms on the $k = 1, 2, \ldots, 39$ best-performing teams on the training set. Next, we chose $k$ to be the value that achieved the best performance on the validation set; this step identified $k = 14$ for the GBT models and $k = 2$ for the RF models on the murmur detection task and $k = 14$ for the GBT models and $k = 4$ for the RF models on the clinical outcome identification task. Finally, we evaluated the resulting voting algorithms on the test set.

The GBT and RF voting algorithms performed slightly better than the highest-ranked entry for the murmur detection task (weighted accuracy metric of 0.790 and 0.789, respectively vs. 0.780; see Table 5) and performed slightly worse than the highest-ranked entries for the clinical outcome identification task (cost of 11357 and 11687, respectively vs. 11144; see Table 6). In each case, the voting algorithms had comparable performance to the highest ranked individual algorithms, but they did not significantly outperform them in either task by any of the traditional or novel scoring metrics that we considered.

## Discussion

Cardiac auscultation is one of the most cost-effective tools for helping clinicians to identify heart murmurs, and the CirCor Digiscope dataset enriches our understanding cardiac auscultation within a resource-constrained environment. However, despite the novelty and value of this dataset, it, like every dataset, has limitations.

No ages were available for the pregnant individuals in the CirCor DigiScope dataset. It was unclear to the teams if the pregnant individuals belonged to the pediatric age group of the rest of the patients, or if they had a different set of exclusion criteria from the other patients, potentially limiting the performance and appropriateness of models that use this dataset on pregnant individuals.

There was a single human annotator for labeling the heart murmurs; we did not have information about the number of human annotators for labeling the clinical outcomes of abnormal or normal heart function. We expect disagreements between annotators, and several annotators are often needed to produce a single, consistent annotation of a health record. Even several annotators may produce disparate annotations [59]. It is likely that different or more annotators would result in different annotations.

Some issues, such as background noises in the recordings and the relatively high heart rates of children, are inherent to the data or to the collection of the data. To preserve the realism of the problem and the diversity of the approaches, we did not remove these issues, enumerate

them as potential difficulties, or suggest how to address them; many of the solutions employ different strategies to address different problems with the data.

Some heart murmurs are pathological, indicating a physiological problem that requires monitoring and/or intervention. Other heart murmurs are innocent, with sounds resulting from normal blood flow, and do not require treatment. However, the human annotator did not identify which heart murmurs were pathological or innocent; the clinical outcomes, which reflect a full examination of the patient, are the strongest evidence in the dataset for characterizing the severity of any heart murmurs or underlying cardiac disease. Therefore, we only evaluated the ability of algorithms to identify heart murmurs, not their ability to identify pathological heart murmurs or differentiate between pathological and innocent heart murmurs, but this task is still valuable for screening for cardiovascular disease.

The definition of a task and the choice of evaluation metrics for quantifying an algorithm's performance on the task affect the actual and perceived clinical relevance of the algorithms [14]. The definitions of the Challenge evaluation metrics (1) and (8) are attempts to make the evaluation metrics, and therefore the algorithms developed for these metrics, more clinically relevant. The correlation in performance between the traditional and Challenge metrics demonstrates that methods that perform better by one metric tended to perform better by another metric (at least for the murmur detection task), but the best-scoring method by one metric was often different from the best best-scoring method by another metric, motivating the careful consideration of metrics; see Figs 6 and 7. We also recognize that these metrics are imperfect descriptions of clinical needs; in particular, healthcare cost can be a poor proxy for health needs [13, 14].

The relatively poor performance of the methods by all metrics on the clinical outcome identification task suggests the difficulty of performing this task using the PCG recordings alone; we note that echocardiography is a standard diagnostic tool for assessing cardiac function, which is a more expensive and less accessible modality than phonocardiography.

The voting algorithms had only modest performance improvements, if any, over the individual algorithms. Indeed, the voting algorithms did not use any data-derived features, which could help to provide context by associating different algorithms with different patient subgroups. The features for the voting algorithms included the top $k$ algorithms, even when including a lower-ranked entry with a worse overall score or excluding a higher-ranked entry with a better overall score would improve the performance of the voting algorithm. This experiment only considered GBT and RF models, but other types of models could potentially achieve better performance. The training procedure for the GBT and RF models did not use hyperparameter optimization (beyond the selection of the number $k$ of individual algorithms), limiting their performance as well. Also, unlike the Challenge teams, we had access to the test set, but, like the Challenge teams, our formal training procedure did not use it. Despite these limitations, the ability of the voting algorithms to slightly improve on the performance on the top-ranked algorithms for the murmur detection task, but not the clinical outcome identification task, also suggests the differences in difficulty between the two related tasks considered by the Challenge.

## Conclusions

The George B. Moody PhysioNet Challenge explored the potential for algorithmic pre-screening for heart murmurs and abnormal cardiac function in resource-constrained environments. We invited the Challenge participants to design working, open-source algorithms for identifying heart murmurs and clinical outcomes from PCG recordings, resulting in 53 working implementations of different algorithmic approaches for interpreting PCGs. A voting

approach to combining the diverse approaches resulted in a superior performance over the individual algorithms for murmur detection but not for clinical outcome identification.

By reducing human screening of patients with normal cardiac function, algorithms can lower healthcare costs and increase the effective screening capacity for patients with abnormal cardiac function. The cost function proposed in this Challenge could be the basis for cost-effective screening that balances both the financial and health burden of correctly or incorrectly diagnosing patients. However, it will be important to optimize the proposed cost function for a given healthcare system or population, since disease prevalence, financial resources, and healthcare resources can differ enormously in different settings.

## Supporting information

**S1 Appendix. Additional scoring metrics.**
(PDF)

**S2 Appendix. Mathematical derivation of the cost-based scoring metric.**
(PDF)

## Acknowledgments

None of the funding entities influenced the design of the Challenge or provided data for the Challenge. The content of this manuscript is solely the responsibility of the authors and does not necessarily represent the official views of the funding entities.

## Author Contributions

**Conceptualization:** Matthew A. Reyna, Andoni Elola, Jorge Oliveira, Francesco Renna, Sandra Mattos, Miguel T. Coimbra, Reza Sameni, Ali Bahrami Rad, Gari D. Clifford.

**Data curation:** Matthew A. Reyna, Andoni Elola, Jorge Oliveira, Francesco Renna, Sandra Mattos, Miguel T. Coimbra, Reza Sameni, Ali Bahrami Rad, Gari D. Clifford.

**Formal analysis:** Matthew A. Reyna, Yashar Kiarashi, Andoni Elola, Francesco Renna, Reza Sameni, Ali Bahrami Rad, Gari D. Clifford.

**Funding acquisition:** Matthew A. Reyna, Sandra Mattos, Miguel T. Coimbra, Reza Sameni, Gari D. Clifford.

**Investigation:** Matthew A. Reyna, Yashar Kiarashi, Sandra Mattos, Miguel T. Coimbra, Reza Sameni, Ali Bahrami Rad, Gari D. Clifford.

**Methodology:** Matthew A. Reyna, Yashar Kiarashi, Andoni Elola, Jorge Oliveira, Francesco Renna, Reza Sameni, Ali Bahrami Rad, Gari D. Clifford.

**Project administration:** Matthew A. Reyna, Sandra Mattos, Miguel T. Coimbra, Reza Sameni, Ali Bahrami Rad, Gari D. Clifford.

**Resources:** Matthew A. Reyna, Jorge Oliveira, Francesco Renna, Annie Gu, Erick A. Perez Alday, Nadi Sadr, Ashish Sharma, Sandra Mattos, Miguel T. Coimbra, Reza Sameni, Ali Bahrami Rad, Gari D. Clifford.

**Software:** Matthew A. Reyna, Andoni Elola, Annie Gu, Reza Sameni, Ali Bahrami Rad, Gari D. Clifford.

**Supervision:** Matthew A. Reyna, Sandra Mattos, Miguel T. Coimbra, Reza Sameni, Ali Bahrami Rad, Gari D. Clifford.

**Validation:** Matthew A. Reyna, Reza Sameni, Gari D. Clifford.

**Visualization:** Matthew A. Reyna, Andoni Elola, Reza Sameni, Gari D. Clifford.

**Writing – original draft:** Matthew A. Reyna, Yashar Kiarashi, Andoni Elola, Jorge Oliveira, Francesco Renna, Reza Sameni, Ali Bahrami Rad, Gari D. Clifford.

**Writing – review & editing:** Matthew A. Reyna, Yashar Kiarashi, Andoni Elola, Jorge Oliveira, Francesco Renna, Jacques Kpodonu, Reza Sameni, Ali Bahrami Rad, Gari D. Clifford.

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
