## [Decision Letter · Decision Letter 0]

19 May 2023

PDIG-D-23-00131

Heart murmur detection from phonocardiogram recordings: The George B. Moody PhysioNet Challenge 2022

PLOS Digital Health

Dear Dr. Reyna,

Thank you for submitting your manuscript to PLOS Digital Health. After careful consideration, we feel that it has merit but does not fully meet PLOS Digital Health's publication criteria as it currently stands. Therefore, we invite you to submit a revised version of the manuscript that addresses the points raised during the review process.

Please submit your revised manuscript within 30 days Jul 18 2023 11:59PM. If you will need more time than this to complete your revisions, please reply to this message or contact the journal office at digitalhealth@plos.org. Please include the following items when submitting your revised manuscript:

We look forward to receiving your revised manuscript.

Kind regards,

Leo Anthony Celi, MD MS MPH

Editor-In-Chief

PLOS Digital Health

Journal Requirements:

2. Please send a completed 'Competing Interests' statement, including any COIs declared by your co-authors. If you have no competing interests to declare, please state "The authors have declared that no competing interests exist". Otherwise please declare all competing interests beginning with twhe statement "I have read the journal's policy and the authors of this manuscript have the following competing interests:"

Additional Editor Comments (if provided):

Reviewers' comments:

Reviewer's Responses to Questions

**Comments to the Author**

1. Does this manuscript meet PLOS Digital Health’s publication criteria? Is the manuscript technically sound, and do the data support the conclusions? The manuscript must describe methodologically and ethically rigorous research with conclusions that are appropriately drawn based on the data presented.

Reviewer #1: Yes

Reviewer #2: Yes

2. Has the statistical analysis been performed appropriately and rigorously?

Reviewer #1: Yes

Reviewer #2: Yes

3. Have the authors made all data underlying the findings in their manuscript fully available (please refer to the Data Availability Statement at the start of the manuscript PDF file)?

Reviewer #1: Yes

Reviewer #2: Yes

4. Is the manuscript presented in an intelligible fashion and written in standard English?

Reviewer #1: Yes

Reviewer #2: Yes

5. Review Comments to the Author

Reviewer #1: The manuscript presents results from the “Heart murmur detection from phonocardiogram recordings: The George B. Moody PhysioNet Challenge 2022”. It is well written and easy to follow. The manuscript uses up to date references; which are all well linked within the document. All reference materials have been provided.

However, the following could be addressed.

- The abstract is great, but consider adding a paragraph about the results (submitted algorithms) from the challenge. The abstract presents the background information; and how the challenge was conducted, but is very silent about the results (which algorithm performed best from the challenge). You might also consider mentioning the aim of this opinion paper in the abstract, to provide a reader with the general picture of the paper.

- The usage of the phrase “This year’s challenge” appears misleading in some paragraphs. May be consider using “The 2022 year’s challenge”; so that it is easy for the reader not to confuse it with the current year.

- The paragraph “while algorithms that did not learn from the training set performed the same or better” on page 12/24 is incomplete.

- The paragraph “requires monitoring and/or intervention” on page 14/24 is incomplete.

- Check paragraph “models and k = 2 for the RF models.” on page 13/24

Reviewer #2: Thank you for the opportunity to review this interesting manuscript. This purpose of the paper was to report findings from The 2022 George B. Moody PhysioNet Challenge (formerly the PhysioNet/Computing in Cardiology Challenge). This challenge invited teams to develop fully automated approaches for detecting heart murmurs and abnormal cardiac function from PCG recordings. The investigators sourced and shared PCG recordings for up to four auscultation locations from a largely pediatric population in Brazil, and teams were asked to identify both heart murmurs and the clinical outcomes of a full diagnostic screening from the recordings. The Challenge explored the diagnostic potential of algorithmic approaches for interpreting PCG recordings. Below are suggestions designed to enhance the readability, clinical understanding and reproducibility of this already well written manuscript. 

The format of the paper made it much easier to review this manuscript – thank you. 

Title – consider adding “Pediatric” to the title. 

Perhaps this was missed by the reviewer, but I did not see key words, are they required?

A limitation of this paper is that the sample was not clearly described. The authors state that the PCG recording were from pediatric patients, yet pregnant individuals were mentioned as having contributed recordings. One might assume if pregnant women were included the PCG recording were obtained from a fetus on the abdomen of the mother? Among pediatric patients one might assume the PCG recordings were obtain from the child’s chest? How might this impact recordings (chest vs abdomen) is this is how the PCG was recorded? 

While it may be outside of the scope of this paper one would also imagine that PCG recordings are impacted by heart rate, which in pediatric patients (in your study you included Neonate (birth to 27 days), Infant (28 days to 1 year), Child (1 to 11 years), Adolescent (12 to 18 years), or fetus, in the case of pregnant individuals, which would be very challenging since heart rate is so fast in younger children. Was heart rate considered or accounted for? Again, this may not be appropriate to discuss in this paper, but could be mentioned. 

In the Introduction, second sentence the term “technology is used. Yet a stethoscope is a device, which may be a more appropriate term.

Introduction second paragraph – the statements regarding “within the valves” is not entirely accurate. Suggest using “around,” rather than “within” as blood flow is disrupted around an incompetent valve. The authors may want to delete the statement “…..and by the turbulence of blood flow within the valves” in the first sentence and end it after “….cardiac cycle.” Then make the suggested adjustment in the next sentence with refers to the pathology. 

Figure 1 – the AV and PV locations are both at the location of the second intercostal space yet the image shows them slightly off, suggesting that the PV is in the third intercostal space. 

Page 4 second paragraph – Consider adding (described below) to the sentence “Each patient’s PCGs and clinical notes were also annotated for murmurs.” This addition would help the reader understand that this description will be discussed further down in the paper.

Can “resource constrained” be defined i.e., rural vs urban, socioeconomic etc. this was not entirely clear. 

Page 6 sentence that reads “………………important issue in the Section .” seem to be missing the section #. 

For the text description for Figures 3 and 4 consider bolding the text for “per-patient” and “per-screening” to add clarity to the description.

6. PLOS authors have the option to publish the peer review history of their article (what does this mean?). If published, this will include your full peer review and any attached files.

**Do you want your identity to be public for this peer review?** For information about this choice, including consent withdrawal, please see our Privacy Policy.

Reviewer #1: Yes: Dr. Wasswa William | Mbarara University of Science and Technology

Reviewer #2: No

---

## [Editor Report · Decision Letter 1]

29 Jun 2023

PDIG-D-23-00131R1

Heart murmur detection from phonocardiogram recordings: The George B. Moody PhysioNet Challenge 2022

PLOS Digital Health

Dear Dr. Reyna,

Thank you for submitting your manuscript to PLOS Digital Health. After careful consideration, we feel that it has merit but does not fully meet PLOS Digital Health's publication criteria as it currently stands. Therefore, we invite you to submit a revised version of the manuscript that addresses the points raised during the review process.

Please submit your revised manuscript within 30 days Jul 29 2023 11:59PM. If you will need more time than this to complete your revisions, please reply to this message or contact the journal office at digitalhealth@plos.org. Please include the following items when submitting your revised manuscript:

We look forward to receiving your revised manuscript.

Kind regards,

Leo Anthony Celi, MD MS MPH

Editor-In-Chief

PLOS Digital Health

Journal Requirements:

2. Please send a completed 'Competing Interests' statement, including any COIs declared by your co-authors. If you have no competing interests to declare, please state "The authors have declared that no competing interests exist". Otherwise please declare all competing interests beginning with twhe statement "I have read the journal's policy and the authors of this manuscript have the following competing interests:"
---

## [Editor Report · Decision Letter 2]

10 Jul 2023

Heart murmur detection from phonocardiogram recordings: The George B. Moody PhysioNet Challenge 2022

PDIG-D-23-00131R2

Dear Prof. Reyna,

We are pleased to inform you that your manuscript 'Heart murmur detection from phonocardiogram recordings: The George B. Moody PhysioNet Challenge 2022' has been provisionally accepted for publication in PLOS Digital Health.

Best regards,

Leo Anthony Celi, MD MS MPH

Editor-In-Chief

PLOS Digital Health